# Long-term potentiation and depression regulatory microRNAs were highlighted in Bisphenol A induced learning and memory impairment by microRNA sequencing and bioinformatics analysis

Mengxin Luo[1,2☯], Ling Li[3☯], Muyao Ding[1☯], Yurong Niu[3], Xuezhu Xu[3], Xiaoxia Shi[1], Ning Shan[2], Zewen Qiu[4]*, Fengyuan Piao[1,5]*, Cong Zhang[1]*

1 School of Public Health, Dalian Medical University, Dalian, China, 2 Panjin Liaoyou Gem Flower Hospital, Panjin, China, 3 Department of Dermatology, The Second Hospital of Dalian Medical University, Dalian, China, 4 Laboratory Animal Center, Dalian Medical University, Dalian, China, 5 Department of Nephrology, Affiliated Zhongshan Hospital of Dalian University, Dalian, China

☯ These authors contributed equally to this work.
* 917640792@qq.com (ZQ); piao_fy_dy@163.com (FP); congzhang1203@hotmail.com (CZ)

## Abstract

The mechanisms of Bisphenol A (BPA) induced learning and memory impairment have still not been fully elucidated. MicroRNAs (miRNAs) are endogenous non-coding small RNA molecules involved in the process of toxicant-induced neurotoxicity. To investigate the role of miRNAs in BPA-induced learning and memory impairment, we analyzed the impacts of BPA on miRNA expression profile by high-throughput sequencing in mice hippocampus. Results showed that mice treated with BPA displayed impairments of spatial learning and memory and changes in the expression of miRNAs in the hippocampus. Seventeen miRNAs were significantly differentially expressed after BPA exposure, of these, 13 and 4 miRNAs were up- and downregulated, respectively. Bioinformatic analysis of Gene Ontology (GO) and pathway suggests that BPA exposure significantly triggered transcriptional changes of miRNAs associated with learning and memory; the top five affected pathways involved in impairment of learning and memory are: 1) Long-term depression (LTD); 2) Thyroid hormone synthesis; 3) GnRH signaling pathway; 4) Long-term potentiation (LTP); 5) Serotonergic synapse. Eight BPA-responsive differentially expressed miRNAs regulating LTP and LTD were further screened to validate the miRNA sequencing data using Real-Time PCR. The deregulation expression levels of proteins of five target genes (CaMKII, MEK1/2, IP3R, AMPAR1 and PLCβ4) were investigated via western blot, for further verifying the results of gene target analysis. Our results showed that LTP and LTD related miRNAs and their targets could contribute to BPA-induced impairment of learning and memory. This study provides valuable information for novel miRNA biomarkers to detect changes in impairment of learning and memory induced by BPA exposure.

**Data Availability Statement:** All relevant data are within the manuscript and its Supporting Information files.

**Funding:** This work was supported by the National Natural Science Foundation of China though a grant awarded to FP (No. 81773402). This work was also supported by the Liaoning Provincial Science and Technology Department Project through a grant awarded to XX (No.2021JH1/10400051).

**Competing interests:** The authors have declared that no competing interests exist.

## Introduction

Bisphenol A (BPA) is an important industrial chemical used extensively worldwide in the production of polycarbonate plastics and epoxy resins [1]. In 2015, the global volume consumption of BPA was estimated at 7.7 million metric tons, and could reach 10.6 million metric tons by 2022 with a compound annual growth rate of 4.8% [2–4]. BPA can seep into our foods and the environment during the manufacturing process or daily use [5]. Humans might be exposed to BPA from fetal to adult stages in various ways including water, air, soil environment, food contamination, etc [6]. The exposure routes of BPA could be oral intake, inhalation or dermal contact [7, 8]. A certain amount of studies performed in cellular cultures, rodents, and humans suggest that BPA overexposure exerts deleterious effects by different mechanisms [9]. BPA works as an agonist on estrogen receptors and antagonist on androgen receptors due to its capability to bind classical nuclear or genomic estrogen receptors [10, 11]. BPA can impair male reproductive function via damaging sperm DNA, decreasing testosterone levels and reducing semen quality [12]. In addition, several studies demonstrated that BPA exposure plays adverse health effects on liver function, increases risk of cardiovascular disease, affects glucose metabolism, disturbs immune function and induces several tumors through binding different receptors, modulating transcription factors, or inducing epigenetic changes [13–16]. It was also found that BPA exposure during embryonic and infant periods may exert toxic impacts on a series of physiological processes such as development on nervous system and brain morphology [17, 18]. In addition, increasing evidence indicated that neurotoxic effects of children/adolescent animals could be induced by BPA exposure [19, 20]. Therefore, more studies on the alterations of behaviors due to the children/adolescent BPA exposure and the involved mechanisms are necessary.

Increasing evidence suggested that epigenetic alterations, including changes in histone modifications, DNA methylation and the expression of non-coding RNAs could be crucial regulators in the long-term effects of environmental toxicants [21, 22]. MicroRNAs (miRNAs), a short non-coding RNAs, regulate translation by binding to the 3′ untranslated region (UTR) of mRNAs [23]. miRNAs play crucial roles in almost all fundamental biological and metabolic processes [24]. Of all the mammalian miRNAs identified to date, more than 50% of them are expressed in the brain [25, 26]. Recent studies have addressed miRNAs to be involved in various central nervous system's pathologies [27, 28]. Specifically, miRNAs play important roles in the development and progression of neurotoxicity induced by environment stress [29, 30]. Butler et al. reported that miRNA expression change in brain was critical in impairment of BPA-related changes in social-communication behaviors [31]. Kaur et al. demonstrated that altered miRNA levels were associated with BPA-induced anxiety and stereotypical behaviors [32].

Given the considering that miRNAs expression alteration is an important mediator of the BPA-induced impairment of learning and memory, in this study, we exposed young male mice to BPA in drinking water for 8 weeks and investigated the impact of BPA exposure on the miRNA expression profile of the hippocampus. The DIANA-miRPath v3.0 database was used for bioinformatic analysis. Real-Time PCR and western blot were performed for validation of miRNA expression and target genes' protein levels. Our data would be valuable for revealing novel miRNA biomarkers and possible mechanisms related to impairment of learning and memory induced by BPA exposure.

## Methods and materials

Animal handling and procedures used in this study were approved by the Institutional Animal Care and Use Committee of Dalian Medical University (AEE21022). Housing of animals was

in agreement with the guidelines of the Animal Care and Use Committee of Dalian Medical University. All procedures were performed under strict accordance with National Institute of Health Guide for Care and Use of Laboratory Animals, and all efforts were made to minimize suffering.

## The treatment of animals

Eighty Kunming (KM) mice (Male, aged 4 weeks) weighing 22±2 g were obtained from the Experimental Animal Center of Dalian Medical University. They were housed five per cage under standard conditions, with a 12 h dark-light cycle (lights on at 7:00 a.m.) at 18–22˚C and 50% humidity and were maintained on a standard diet with water available ad libitum. All mice were randomly assigned to four groups (n = 20 in each group), including the control group and three BPA exposure groups, according to their body weight. Mice in the four groups were received the following dosages of BPA in the drinking water: 0, 0.05, 0.5, and 5 mg/kg body weight. All treatments were continued for 8 weeks. The volumes of water consumed were measured every 2 days, the weights of the mice were measured every 1 week, and there were no significant differences in water consumption and the body weights of the mice between the BPA-exposed and control groups (Fig 1).

After 8 weeks of BPA treatment, the test of learning and memory ability was performed in all mice from each group (n = 15–20 in each group). Then, mice were euthanized and hippo-campus tissues were collected. The hippocampus tissues of three mice in control group and 5mg/mg BPA group were used to construct miRNA expression profile by miRNA sequencing, the hippocampus tissues of other six mice in each group were used to Real-Time PCR and western blot analysis.

The concentrations of BPA exposure were set based on the Reference Dose ($R_fD$) set by the Environmental Protection Agency (0.05 mg/kg body weight, the low BPA-exposed group), 10 times the $R_fD$ (0.5 mg/kg body weight, the medium BPA-exposed group) and 100 times the $R_fD$ (5 mg/kg, the high BPA-exposed group) [33].

## Tests of animal learning and memory ability

Mice from each group were tested for their learning and memory ability using the MWM test as described in our previous research [34]. The device is a circular pool with a diameter of 100 cm, and is filled with water at 23 ± 2˚C (depth of water: 40 cm). An escape platform (diameter: 10 cm) was submerged 1 cm below the water surface, was placed in the middle of one quadrant (target quadrant) during the spatial navigation test. Spatial visual cues were black and white geometric figures including triangle, square, circle and diamond placed around the pool in each quadrant. All tests were performed between 15:00 and 22:00, and the water was changed

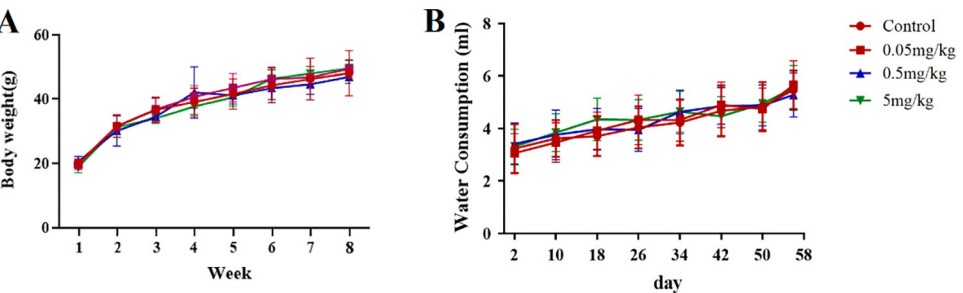

**Fig 1.** Body weight (a) and water consumption (b) of mice (n = 20). Results are presented as mean ± SEM.

to fresh water daily. The water maze was surrounded by shading curtains for avoiding the interference of light on image acquisition. Briefly, the test included the spatial acquisition phase and probe trial [35, 36]. In spatial acquisition phase, mice were trained for 5 consecutive days. Daily training was starting at different quadrants of the pool in a predetermined pseudo-random order. The time spent to reach the platform (escape latency) within 60 s was recorded as acquisition latency. If the mouse failed to reach the platform within 60 s, it was gently guided onto the platform for 30 s and assigned a latency of 60 s. The escape latency to identify the hidden platform was measured.

On the sixth day, the mice were given a spatial probe test, in which the platform was taken away, and each mouse was placed at one point. The mouse was allowed to navigate freely in the pool for 60 s. The time of swimming in the target quadrant, which had previously contained the hidden platform, was recorded as an indicator of memory retention. The swim paths of mice were recorded and the crossings in the target quadrant were calculated by a smart video tracing system (NoldusEtho Vision system, version 5, Everett, WA, USA).

## miRNA library construction and sequencing

Total RNA was extracted from hippocampal tissues by using RNAiso Plus according to the manufacturer's instruction (Takara, Japan) and then quantified with the NanoPhotometer® spectrophotometer (IMPLEN, CA, USA). Only RNA samples with an A260/A280 of 1.8–2.2 were employed for reverse transcription. RNA concentration was measured using Qubit® RNA Assay Kit in Qubit® 2.0 Flurometer (Life Technologies, CA, USA). RNA integrity was assessed using the RNA Nano 6000 Assay Kit of the Agilent Bioanalyzer 2100 system (Agilent Technologies, CA, USA). 3 μg total RNA per sample was used for the small RNA library. Sequencing libraries were generated with NEBNext® Multiplex Small RNA Library Prep Set for Illumina® (NEB, USA) following manufacturer's recommendations. The quality of library was evaluated by the Agilent Bioanalyzer 2100 system using DNA High Sensitivity Chips. The clustering of the index-coded samples was performed on a cBot Cluster Generation System using HiSeq Rapid Duo cBot Sample Loading Kit (Illumia) according to the manufacturer's instructions. After cluster generation, the library preparations were sequenced on an Illumina Hiseq 2500 platform and 50 bp single-end reads were generated.

## Bioinformatic evaluation

After miRNA reads were counted and normalized, fold change (FC) between the control and BPA-treated mice was calculated by R program (V3.6.2). The genes whose sum of readcount values of the two groups was less than 10 were filtered out. We selected the differentially expressed miRNAs according to the log2 (FC) and $p$ value threshold. |log2 (FC) |$\geq$ 1 and $p$ value < 0.05 was considered as significant difference. The scatter map and clustering heat map of differential expression miRNAs were presented by R program (V3.6.2) and Python program.

After the differentially expressed miRNAs were screened out, bioinformatic evaluation was carried out for identifying potential functions and prediction of target genes as described previously [30]. The functional enrichment analysis was carried out by DIANA-miRPath v3.0. The Gene Ontology (GO) terms and KEGG pathway analysis were provided to obtain useful information regarding the functions of the altered miRNAs [37–39]. The $p$-value reflects the significance of GO term enrichment and the pathways correlated to the conditions (The threshold of $p$-values corrected by false discovery rate (FDR) is 0.05).

The biological significance of altered miRNA expression is intimately associated with their gene targets. Potential target genes of all the differentially expressed miRNAs were predicted

from data in the databases: miRDB, miRanda, miRWalk, TargetScan, DIANA-mirPath, and miRNA.org. Then, the results intersected from at least two different programs were retained as the final set of target genes.

## Real-Time PCR

One microgram of total RNA was reverse transcribed to cDNA using miRNA Reverse Transcription reagent (Takara, Japan). Quantitative Real-Time PCR was performed with a SYBR Green PCR kit (Takara, Japan) using the TP800 Real-Time PCR Detection System (Takara, Japan). The Bulge-Loop miRNA primers for the selected miRNAs and U6 miRNA were designed and purchased from RiboBio (Guangzhou, China). The U6 small nuclear RNA was used as endogenous control. The primers for the selected genes are shown in S1 Table. The reaction conditions were set as follows: initial denaturation at 94°C for 2 min, followed by 5 cycles of 94°C for 30 s, 55°C for 30 s and 60°C for 30 s. The data were analyzed using the $2^{-\triangle\triangle CT}$ method.

## Western blot analysis

Western blot analysis was performed to detect the protein expression of CaMKII, MEK1/2, AMPAR1, IP3R and PLCβ4. GAPDH was used as a control. Briefly, samples were homogenized in ice-cold RIPA Tissue Protein Extraction Reagent (Biyuntian, China) supplemented with 1% proteinase inhibitor mix and incubated at 4°C for 1h. After incubation, debris was removed via centrifugation at 12,000×g for 15 min at 4°C, and the lysates were stored at -80°C until being used. The total protein concentration in the lysates was determined using a BCA protein assay kit (Biyuntian, China). The samples employed for western blot contained 50 μg of protein from tissues in each lane. The proteins were mixed with an equal volume of SDS-PAGE loading buffer, separated via SDS-PAGE under non-reducing conditions using 10% SDS-PAGE gels and electrotransferred to Hybond-P polyvinylidene fluoride membranes (Millipore, France). The membranes were blocked with blocking buffer containing defatted milk powder for 1 h and incubated overnight at 4°C with 1μg/ml of anti-rabbit CaMKII (1:1000, Abcam, catalog #ab181052), MEK1/2 (1:1000, Abcam, catalog #ab178876), IP3R (1:1000, Abcam, catalog #ab264281), AMPAR1 (1:1000, Cell signaling Technology, catalog #13185S) and PLCβ4 (1:1000, Absin, catalog #abs132943) antibodies. The membrane was washed three times with Tris-buffered saline containing 0.05% Tween-20 (TBST) for 15 min and then incubated at room temperature for 1 h with horseradish peroxidase-conjugated goat anti-rabbit IgG (1:3,000, Sigma-Aldrich, catalog #A0545). The resultant signals were visualized using an enhanced ECL chemiluminescence kit and quantified densitometrically using a UVP BioSpectrum Multispectral Imaging System (Ultra-Violet Products Ltd, Upland, CA). Finally, the band intensity of the target proteins was read using ImageJ software (Bethesda, MD, U.S. A.) against the control sample, GAPDH staining was performed in stripped membranes used to identify the target protein's antibodies, and normalization was performed by the ratio from band intensity of target proteins against GAPDH.

## Statistical analysis

Prior to analyses, we confirmed data normality and homogeneity of variances by D'Agostino–Pearson and Barlett's tests, respectively. Difference among each group was analyzed by one-way analysis of variance (ANOVA), followed by the Tukey's multiple-comparison tests. All data set fellows a normal distribution, and are presented as the mean ± standard error of mean (SEM). In all instances a $p$-value of less than 0.05 was considered to be statistically significant. All data were analyzed with SPSS 23.0 for Windows.

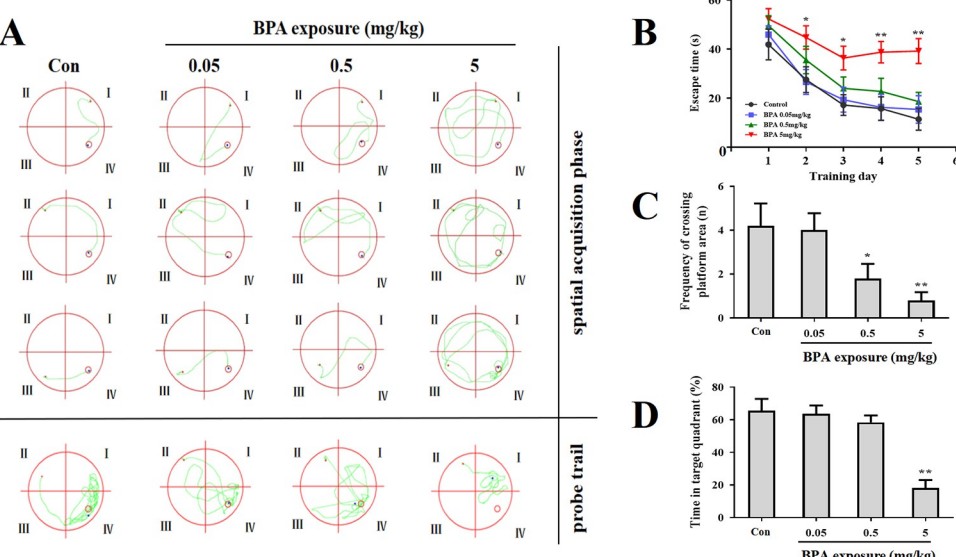

**Fig 2. Impairment of learning and memory in BPA-treated mice in the MWM (n = 15–20).** (A) Representative swimming routes of each group during the spatial acquisition phase (the 5th day) and probe trial in the MWM. The representative swimming routes belong to the same animal in each group. In spatial acquisition phase, the representative swimming routes of trail 1–3 are the paths for finding the platform in target quadrant from quadrant I, II or III, respectively. In the probe trial, the representative swimming routes of trail 4 are from the quadrant II to quadrant IV for navigating freely 60 s. (B) In spatial acquisition phase, the time spent to find the hidden platform of mice after exposure to BPA is shown. The data presented is the daily average of trainings in four different quadrants. In the probe trial, the total number of crossings over the platform (C) and the time spent (D) in the target quadrant of mice after exposure to BPA were shown. Results are presented as mean ± SEM. $^*P < 0.05$, $^{**}P < 0.01$ compared with control group.

## Results

### BPA induces impairment of learning and memory

Though MWM test, we found that BPA exposure impaired the mice learning and memory abilities. The representative swimming routes of each group during the spatial acquisition phase (the 5th day) and probe trial were shown (Fig 2A). In the spatial acquisition phase, no significant differences were observed in 0.05 and 0.5 mg/kg BPA-treated mice compared with the control group. 5 mg/kg BPA-treated mice spent a longer time to find the hidden platform than the control mice from the second day (Fig 2B). In the probe test, compared with the control group, the mice in 5 mg/kg BPA-exposed group crossed the platform fewer times (Fig 2C), and stayed shorter in the target quadrant (Fig 2D).

### BPA promotes an alteration of miRNA expression profile in the mice hippocampus

Since BPA exposure at 5 mg/kg impairs learning and memory ability of mice, we investigated the possible expression alterations of miRNAs between the BPA treatment and the control groups in mouse hippocampus. We screened out miRNAs whose relative expression levels changed more than 2 fold between the BPA-treated and control groups. In Fig 3, it was shown that the expression of 17 miRNAs were significantly changed after the BPA treatment ($p < 0.05$, fold change ≥ 2), of these, 13 were upregulated and 4 were downregulated (Table 1). Variations in the expression of miRNAs between the BPA and control group was exhibited with the clustering heatmap (Fig 4).

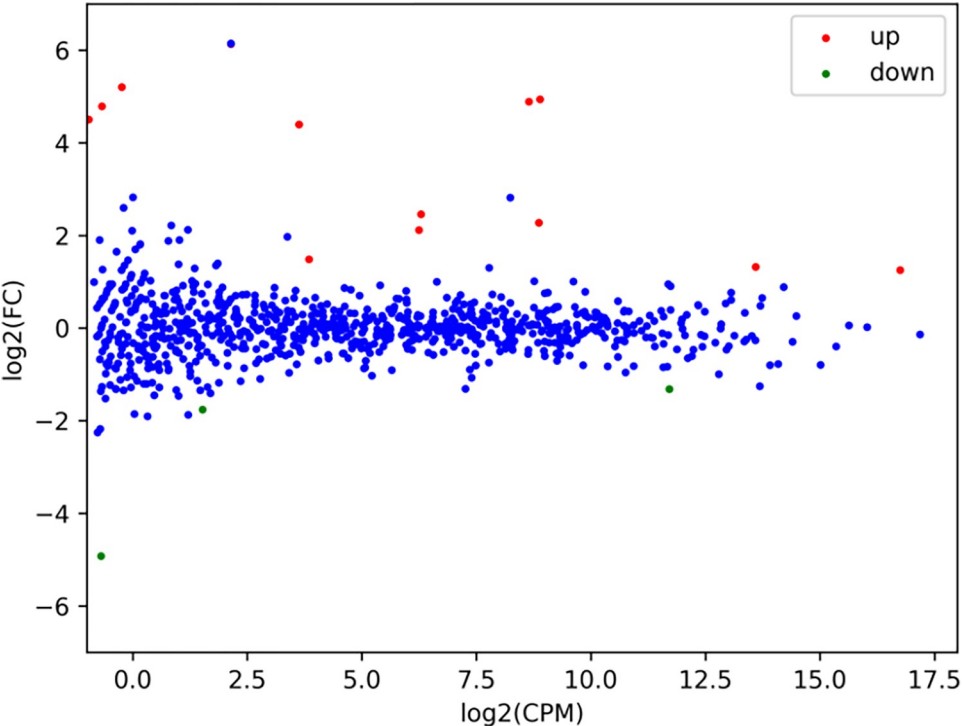

**Fig 3. Scatter plot of differentially expressed miRNAs in hippocampus between the BPA treatment group and the control group (n = 3).** Blue dots represent miRNAs with no differential expression; Red and green dots indicate up and downregulation of miRNA, respectively, relative to the control (log2-scaled, $p < 0.05$).

**Table 1. miRNAs with statistical differences in expression between the BPA treatment and control groups.**

| miRNA | Mean±SEM | | *p*-value |
|---|---|---|---|
| | **CON** | **BPA** | |
| mmu-miR-615-3p | 0.286±0.048 | 4.127±0.259 | <0.001 |
| mmu-miR-10b-3p | 0.303±0.075 | 6.275±0.943 | 0.003 |
| mmu-miR-3074-5p | 4.711±0.373 | 1.561±0.261 | 0.002 |
| mmu-miR-24-3p | 4.968±0.341 | 2.201±0.202 | 0.002 |
| mmu-miR-182-5p | 0.242±0.059 | 1.459±0.266 | 0.011 |
| mmu-miR-10a-5p | 0.416±0.183 | 6.083±0.504 | <0.001 |
| mmu-miR-7039-5p | 0.758±0.243 | 7.494±0.389 | <0.001 |
| mmu-miR-125a-3p | 1.047±0.108 | 2.696±0.265 | 0.005 |
| mmu-miR-7021-5p | 4.318±0.536 | 0.571±0.169 | 0.003 |
| mmu-miR-6901-5p | 0.184±0.068 | 1.808±0.169 | 0.001 |
| mmu-miR-96-5p | 1.688±0.182 | 4.513±0.424 | 0.004 |
| mmu-miR-10b-5p | 0.367±0.099 | 4.729±0.749 | 0.004 |
| mmu-miR-193a-3p | 5.198±0.552 | 0.872±0.232 | 0.002 |
| mmu-let-7c-5p | 1.107±0.327 | 2.134±0.121 | 0.042 |
| mmu-miR-7071-3p | 0.147±0.032 | 1.429±0.118 | <0.001 |
| mmu-miR-495-3p | 0.958±0.168 | 4.705±0.331 | 0.001 |
| mmu-miR-183-5p | 0.483±0.057 | 2.983±0.358 | 0.002 |

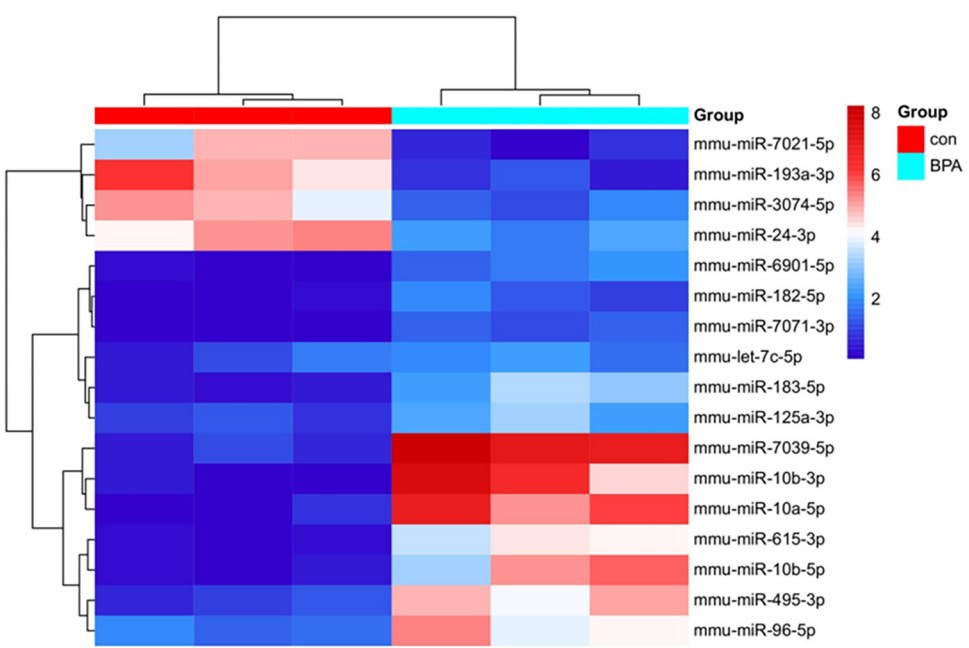

**Fig 4. Heat map depicting the clustering of significantly differentially expressed miRNAs after BPA exposure.**
Red represents high relative expression, and blue represents low relative expression.

## Potential signaling pathways for BPA-affected dysregulated miRNA

To gain a better understanding about the function of the differentially expressed miRNAs, the Gene Ontology annotations and their molecular pathways were summarized via mirPath v.3 analysis [30]. The top 10 significantly changed GO terms (GOTERM BP FAT) biological process were highlighted in Table 2. The term "learning and memory" ($p$ = 1.1E-4) showed the significant change.

Furthermore, 8 KEGG pathways were significantly enriched by the dysregulated miRNAs (Table 3): including Long-term depression (LTD), Thyroid hormone synthesis, GnRH signaling pathway, Long-term potentiation (LTP), Serotonergic synapse, Neurotrophin signaling pathway, Glutamatergic synapse, MAPK Signaling Pathway.

**Table 2. The top 10 GO terms significantly changed in biological process as seen in mice hippocampus after BPA exposure.**

| Term | $p$-Value | Ontology |
|---|---|---|
| Protein phosphorylation | 8.7E-5 | Biological process |
| Signal transduction | 9.8E-5 | Biological process |
| Learning or memory | 1.1E-4 | Biological process |
| Learning | 1.3E-4 | Biological process |
| Chemical synaptic transmission | 1.4E-4 | Biological process |
| Regulation of synaptic plasticity | 1.7E-4 | Biological process |
| MAPK cascade | 2.9E-4 | Biological process |
| Neurological system process | 3.7E-4 | Biological process |
| Negative regulation of neuron apoptotic process | 6.6E-4 | Biological process |
| Locomotory behavior | 8.6E-4 | Biological process |

**Table 3. KEGG pathway analysis of the differentially expressed miRNAs between BPA and control groups.**

| Pathway term | p-Value | Differentially expressed miRNA |
|---|---|---|
| Long-term depression | 5.54E-05 | 7 |
| Thyroid hormone synthesis | 0.00012 | 7 |
| GnRH signaling pathway | 0.006163717 | 10 |
| Long-term potentiation | 0.0111166809095 | 8 |
| Serotonergic synapse | 0.0245688634582 | 10 |
| Neurotrophin signaling pathway | 0.0300138847333 | 11 |
| Glutamatergic synapse | 0.0348900671113 | 10 |
| MAPK signaling pathway | 0.045426897 | 12 |

## BPA exposure alters expression of LTP and LTD related miRNAs

As the top significantly affected pathway after BPA exposure, LTD is a long-lasting decrease in synaptic strength, together with long-lasting increase known as LTP, play crucial roles in the cellular and molecular mechanisms by which memories are formed and stored [40, 41]. Therefore, we focused on the miRNAs related to LTP and LTD.

In Table 4, it was shown that 8 differentially expressed miRNAs were associated with LTP, and 7 were related to LTD. Seven miRNAs play roles in both LTP and LTD, including miR-24-3p, miR-182-5p, miR-96-5p, miR-183-5p, miR-193a-3p, miR-125a-3p and miR-10b-3p; miR-10b-5p only related to LTP. After BPA exposure, miR-10b-3p, miR-182-5p, miR-96-5p, miR-183-5p, miR-10b-5p and miR-125a-3p were upregulated, whereas miR-24-3p and miR-193a-3p were downregulated in BPA group compared with control. We developed Real-Time PCR to confirm the results of miRNA sequencing, the expression levels of these 8 miRNAs displayed a similar regulation trend to the gene sequencing results (Fig 5).

## BPA affected regulation of LTP and LTD regulatory miRNAs' targets involved in neurotoxicity

We set out to further investigate the function of these 8 LTP and LTD regulatory miRNAs by miRNAs-target predictional algorithms, including miRanda, miRWalk, miRDB, TargetScan, miRNA.org and DIANA-mirPath. For minimizing the number of putative and maybe false positive targets, we select intersections from at least two different databases as miRNA's targets [30, 42]. In Table 5, it was shown the detailed information of LTP and LTD regulatory target genes of differentially expressed miRNAs.

Full name of miRNAs' target genes shown in S2 Table.

The functional regulation pathway from BPA exposure to impairment of learning and memory was summarized using LTP and LTD-related miRNAs and their predicted target

**Table 4. The differentially expressed miRNAs regulatory LTP and LTD between BPA and control groups.**

| Pathway term | | Differentially expressed miRNA | Change direction |
|---|---|---|---|
| LTP | LTD | miR-24-3p | Down |
| | | miR-193a-3p | Down |
| | | miR-96-5p | Up |
| | | miR-183-5p | Up |
| | | miR-182-5p | Up |
| | | miR-10b-3p | Up |
| | | miR-125a-3p | Up |
| | | miR-10b-5p | Up |

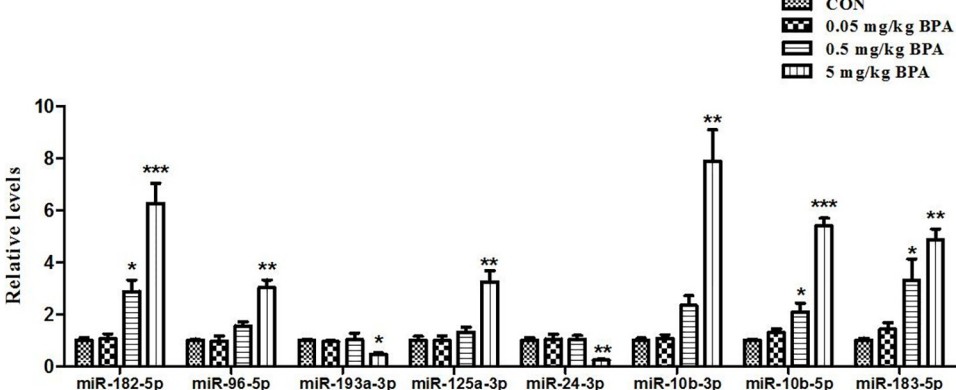

**Fig 5. The differential relative expression of LTP and LTD related miRNAs after BPA treatment in hippocampus validated by Real-Time PCR.** The results are expressed as mean ± SEM (n = 6). $^*p < 0.05$, $^{**}p < 0.01$, $^{***}p < 0.001$ vs the control group.

genes in Fig 6. Some miRNAs can work cooperatively on regulating the target expression. Several miRNAs might regulate more than one target gene in LTP and LTD.

## The confirmation of LTP and LTD regulatory miRNAs' targets

CaMKII, MEK1/2, IP3R, AMPAR1 and PLCβ4 are important proteins in LTP and LTD, the protein levels of them were shown in Fig 7. Compared to the control group, all analyzed protein levels were decreased in the hippocampus of mice exposed to 5 mg/kg BPA; protein levels of CaMKII, MEK1/2, AMPAR1 and PLCβ4 were decreased in group exposed to 0.5 mg/kg BPA; however, protein levels have no significant changes in group exposed to 0.05 mg/kg BPA.

## Discussion

As an endocrine-disrupting chemical, BPA crosses the blood-brain barriers due to its lipophilicity [3] and has suspected roles as a neuroxicant. BPA-induced neurotoxicity has been correlated with enhancement of a neuroinflammatory conditions and oxidative stress, disruption of axon guidance, and other critical cellular processes [43–45]. Although many studies have shown that some miRNAs may contribute to BPA pathological effects [46], their function in BPA induced impairment on learning and memory ability remains to be explored.

In our present study, a brain region involved in learning and memory, hippocampus was selected for analysis the miRNAs expression after BPA exposure. We found that 17 miRNAs

**Table 5. Detail information of LTP and LTD regulatory miRNAs' target gene.**

| Differentially expressed miRNA | LTP and LTD regulatory miRNAs' targets |
|---|---|
| miR-24-3p | Camk2, Rap1a, Rap1b, Gria3, Pla2g4e |
| miR-193a-3p | Kras |
| miR-96-5p | Gnaq, Map2k1, Braf, Kras, Plcb4, Itpr1, Itpr2, Gria1, Camk4, Rps6ka3 |
| miR-183-5p | Plcb4, Rps6ka3 |
| miR-182-5p | Gnaq, Map2k1, Braf, Itpr1, Camk4, Prkacb |
| miR-10b-3p | Grm1, Grm5, Prkca, Camk4, Grin2b, Rap1a, Rps6ka3, Prkacb, Gnai3, Gnaz, Gria3 |
| miR-125a-3p | Gnaq, Braf, Kras, Prkca, Prkcb, Gria1, Rap1a, Gnaz, Gria3 |
| miR-10b-5p | Camk2 |

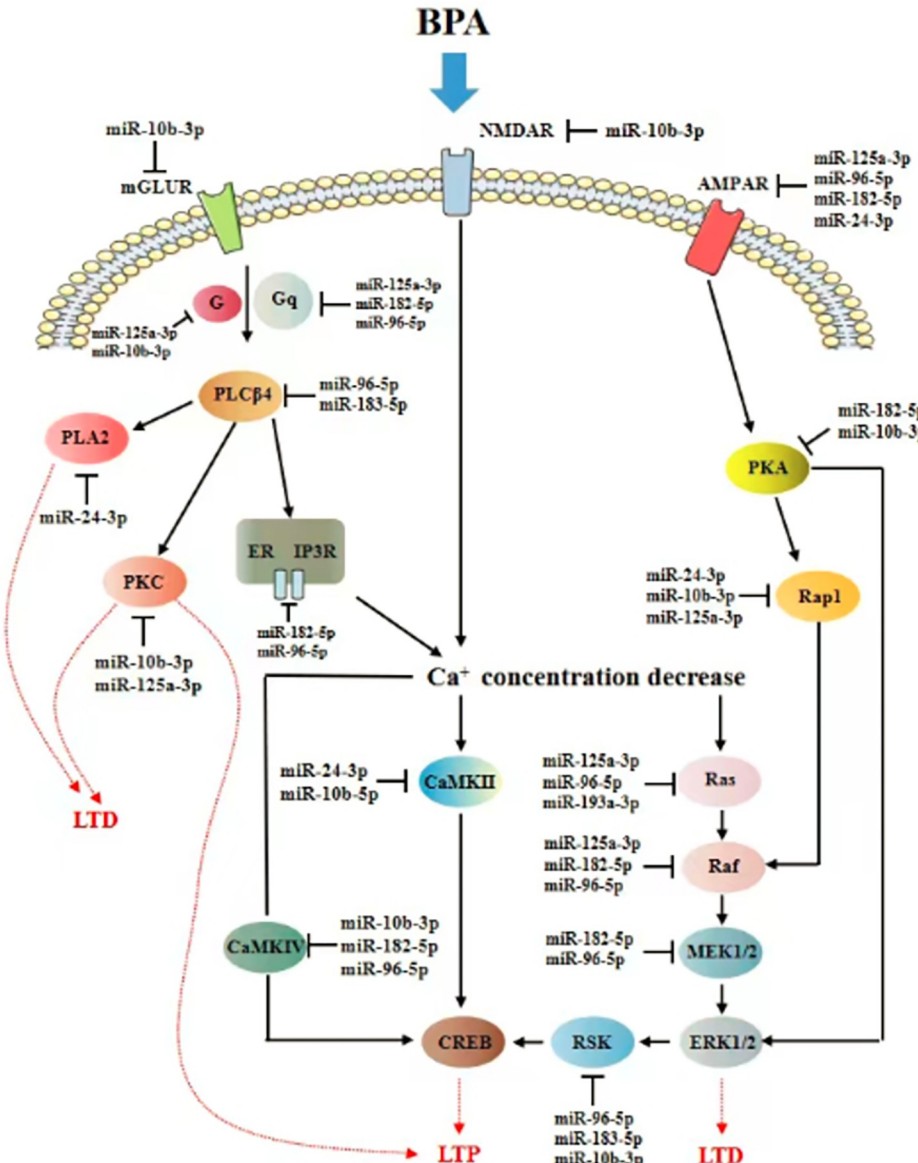

**Fig 6. The miRNAs and their target genes involved in the LTP and LTD.** The pathway is obtained from KEGG pathway and confirmed through NCBI PubMed. The target genes involved in LTP and LTD were predicted with at least two of the following databases: miRDB, miRanda, miRWalk, TargetScan, DIANA-mirPath, and miRNA.org. ER: endoplasmic reticulum; AMPAR: glutamate ionotropic α-amino-3-hydroxy-5-methylisoxazole-4-propionic acid receptor; CaMKII: Calcium/calmodulin-dependent protein kinase II; CaMKIV: Calcium/calmodulin-dependent protein kinase IV; CREB: cAMP responsive element binding protei; ERK1/2: extracellular signal-regulated kinase; G: guanine nucleotide binding protein (G protein); Gq: guanine nucleotide binding protein q polypeptide; IP3R: Inositol 1,4,5-triphosphate receptor; MEK1/2: mitogen-activated protein kinase; mGLUR: metabotropic glutamate receptor; NMDAR: N-methyl-D-aspartate receptor; PLCβ4: Phospholipase Cβ4; PLA2: Phospholipase A2; PKC: protein kinase C; Rap1: RAS related protein 1; PKA: protein kinase A; RSK: ribosomal protein S6 kinase.

whose expression was changed in mice hippocampus by BPA. Bioinformatics analysis was conducted to further study the potential roles of the target genes regulated by these deregulated miRNAs [47]. GO analysis suggested that target genes of these miRNAs were closely associated with regulation of central nervous system structure and function, including several cognitive processes, especially the learning and memory. KEGG pathway analysis revealed the highly

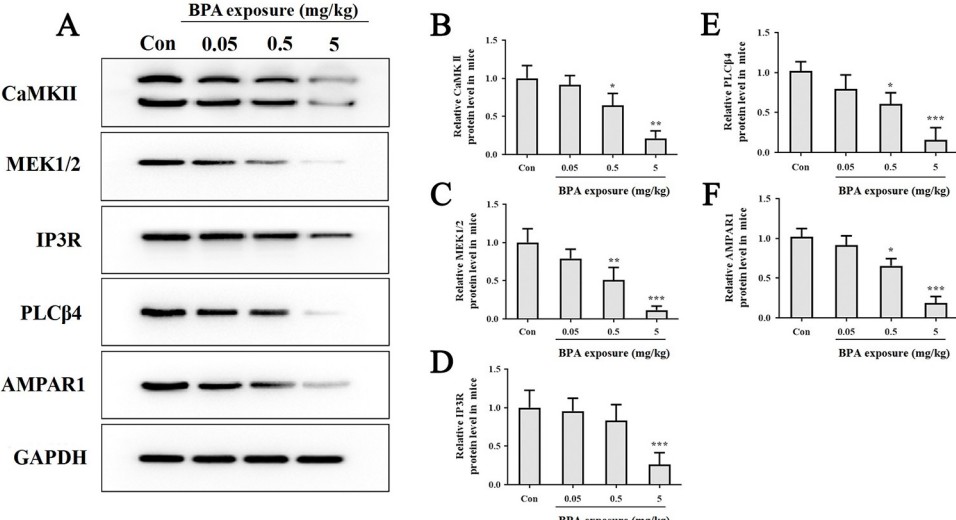

**Fig 7. Western blot analysis of the protein levels of LTP and LTD related miRNAs' target genes in hippocampus.**
The results are expressed as mean ± SEM (n = 6). $^{*}p < 0.05$, $^{**}p < 0.01$, $^{***}p < 0.001$ vs the control group.

enriched learning and memory-associated pathways, interestingly, the top major pathway affected was LTD. It is widely LTP and LTD are two forms of synaptic plasticity in excitatory neurons [48, 49], known as crucial regulators of long-lasting increase and decrease in synaptic strength [49]. Protein synthesis is required for LTP and LTD [50, 51]. Even transcriptional regulation have largely accounted for the molecular mechanisms underlying characteristic changes in long-lasting synaptic plasticity, modulation of mRNA translation attracted more attention and supporting [50]. miRNAs are important small noncoding RNA molecules, they play important regulatory roles in repression of mRNA translation through binding target sites [52]. Our present study demonstrated that levels of 8 miRNAs related to LTP and LTD were affected by BPA treatment, including miR-10b-3p, miR-10b-5p, miR-182-5p, miR-24-3p, miR-96-5p, miR-193a-3p, miR-183-5p, and miR-125a-3p.

Most of these 8 regulated miRNAs were found to be involved in neural system function and neurological diseases. miR-10b-5p was strongly over-expressed in prefrontal cortex of patients with Huntington's disease [53]. miR-182-5p and miR-183-5p have been shown that play a critical role in Attention-deficit/hyperactivity disorder [54]. miR-24-3p and miR-495-3p were shown to participant in pathogenesis of Parkinson's and Alzheimer's disease [55, 56]. Dysfunctions in synaptic plasticity mechanisms can underlie the cognitive deficit in neurological diseases [57, 58]. Several miRNAs like miR-24-3p has been reported changed during LTP, which contributes to long-lasting modification of synaptic function [59]. Previous study found that LTP induction in the CA1 of mice hippocampus was impaired by environmentally low-dose BPA exposure [19]. BPA significantly modulates LTD in the adult rat hippocampus [60]. Our results might indicate that BPA could impair learning and memory ability through regulating these LTP and LTD related miRNAs.

As certral regulatory factors in epigenetic mechanisms, miRNAs have been proved that mediates excitatory synaptic plasticity via regulating local synaptic protein translation [61]. Protein synthesis and abundance of postsynaptic glutamate receptors, structural and signaling factors is crucial for the maintenance of the change in synaptic strength during LTP [62]. miRNAs can target mRNAs by pairing 3′ untranslated regions, inhibit mRNA stability and translation, regulate synaptic protein synthesis, and control synaptic transmission and plasticity [63].

The study by Stefanovic et al., have reported that miR-125a regulated expression of postsynaptic density 95 (PSD-95) [64]. Lee et al. found that miR-188 served as an important modulator for LTP by negatively targeting neuropilin-2 (Nrp-2) [65]. miR-182 regulated synaptic protein synthesis in long-lasting plasticity through Rac1 [66]. In addition, miR-135 acts by complexin-1/2 to manage the NMDAR induced LTD [67]. Since growing evidence reveal that miRNAs play important roles in the effect of toxicants [68], in our study we found that BPA could disturb the expression of miRNAs, then affect the expression of many crucial signals related to LTP and LTD, and finally induce neurotoxicity.

Interestingly, we noticed that genes encoding NMDAR, AMPAR and mGluRs were identified as targets of multiple miRNAs in LTP and LTD. Molecular events underlying the early phase of LTP / LTD include $Ca^{2+}$ influx into a neuron through NMDAR and mGluRs, subsequent direct or indirect activation of CaMKII, and CaMKII-dependent insertion of AMPAR into the post-synaptic membrane [69]. A host of protein kinases, such as PLA2, PLCβ4, PKA, PKC, and MAPK, might be triggered and contributed to LTP in various ways. Our bioinformatics analysis showed that these pathways mentioned above may be all affected by BPA.

The CaMKII family consists of four isoforms (alpha, beta, gamma and delta), of which CaMK2A and CaMK2B are highly expressed in the brain and play roles in both hippocampal plasticity. We showed here that BPA could decrease protein level of CaMKII. Our results were consistent with the previous study of Viberg and Lee et al, in which they found that BPA suppressed the activation of CaMKII in mice hippocampus and cerebral cortex of adult male and female mice [70]. As an essential component of MAPK signal transduction pathway, MEK1/2 (MAPK kinase) is involved in many cellular processes such as differentiation, proliferation, transcription regulation, and development [71]. The over-expression of MEK1/2 can selectively activate the ERK1/2 signaling pathway [72, 73], a pathway involved in BPA-induced impairment of synaptic plasticity [74]. PLCβ4 is an important kinase mediate signals from mGluR1 that are crucial for the modulation of synaptic transmission and plasticity. As a downstream signaling pathway of PLCβ4, IP3R is mediating $Ca^{2+}$ release from the endoplasmic reticulum [75]. The expression of PLCβ4 and IP3R were affected by neurotoxicants [49, 76]. Our work is the first report that expression of PLCβ4 and IP3R can be affected by BPA in brain.

We are finding that the AMPAR is regulating by miR-125a-3p, miR-96-5p, miR-182-5p and miR-24-3p; MEK1/2 and IP3R are the targets of both miR-182-5p and miR-96-5p; CaMKII is regulated by miR-10b-5p and miR-24-3p; PLCβ4 is regulated by miR-96-5p and miR-183-5p. These findings suggest that miRNAs can act cooperatively to regulate target expression in neurons. Our present results indicated that the impairment of learning and memory might be due to, or partly, the results of miRNAs' co-regulation on LTP and LTD.

In addition, some proteins are involved in the regulation of multiple pathways. Such as MEK1/2 protein, belongs to MAPK family, was also identified as targets of the miRNAs in the other pathways, such as MAPK pathway. In the current study, besides the findings on BPA regulate LTP and LTD pathways, some other significantly differentially enriched pathways such as GnRH signaling pathway, Glutamatergic synapse pathway, Serotonergic synapse pathway, Neurotrophin signaling pathway, Glutamatergic synapse pathway, MAPK pathway might take part in the BPA mediated learning and memory impairment. Therefore, further studies are needed to confirm their function in neurotoxicity.

## Conclusion

This is the first study to explore the miRNAs regulation in BPA-induced learning and memory impairment. The results indicate that the differentially expressed miRNAs identified in the

hippocampus could be the targets of BPA, which may play function via LTP and LTD. Our present study not only provides new insights into the pathogenesis of BPA especially linking to impairment of learning and memory, but it also provides us clues for future mechanism exploring.

## Supporting information

**S1 Table. Primers used in Real-Time PCR.**
(DOCX)

**S2 Table. Full name of miRNAs' target genes.**
(DOCX)

## Author Contributions

**Conceptualization:** Cong Zhang.

**Data curation:** Yurong Niu, Xuezhu Xu, Ning Shan.

**Formal analysis:** Xiaoxia Shi.

**Funding acquisition:** Xuezhu Xu, Fengyuan Piao.

**Investigation:** Ling Li, Muyao Ding.

**Methodology:** Mengxin Luo, Zewen Qiu, Fengyuan Piao.

**Writing – review & editing:** Xuezhu Xu, Zewen Qiu, Cong Zhang.

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
