## [Decision Letter · Decision Letter 0]

15 Jun 2022

PONE-D-22-07675Long term potentiation and depression regulatory microRNAs were highlighted in Bisphenol A induced learning and memory impairment by microRNA sequencing and bioinformatics analysisPLOS ONE

Dear Dr. Zhang,

Thank you for submitting your manuscript to PLOS ONE. After careful consideration, we feel that it has merit but does not fully meet PLOS ONE’s publication criteria as it currently stands. Therefore, we invite you to submit a revised version of the manuscript that addresses the points raised during the review process.

I recommend addressing all points raised by the reviewers. Moreover, to increase the implication of the findings, consider the inclusion and discussion of the references below:

https://pubmed.ncbi.nlm.nih.gov/35615589/

https://pubmed.ncbi.nlm.nih.gov/35441941/

https://pubmed.ncbi.nlm.nih.gov/34712129/

https://pubmed.ncbi.nlm.nih.gov/34274347/

https://pubmed.ncbi.nlm.nih.gov/24385256/

We look forward to receiving your revised manuscript.

Kind regards,

Alexandre Hiroaki Kihara, Ph.D.

Academic Editor

PLOS ONE

Journal Requirements:

2.  Please include your tables as part of your main manuscript and remove the individual files. Please note that supplementary tables (should remain/ be uploaded) as separate "supporting information" files

4. PLOS requires an ORCID iD for the corresponding author in Editorial Manager on papers submitted after December 6th, 2016. Please ensure that you have an ORCID iD and that it is validated in Editorial Manager. To do this, go to ‘Update my Information’ (in the upper left-hand corner of the main menu), and click on the Fetch/Validate link next to the ORCID field. This will take you to the ORCID site and allow you to create a new iD or authenticate a pre-existing iD in Editorial Manager. Please see the following video for instructions on linking an ORCID iD to your Editorial Manager account: https://www.youtube.com/watch?v=_xcclfuvtxQ.

6. We note that the grant information you provided in the ‘Funding Information’ and ‘Financial Disclosure’ sections do not match.

7. Thank you for stating the following financial disclosure:

“NO”

8. We noticed you have some minor occurrence of overlapping text with the following previous publication(s), which needs to be addressed:

- https://onlinelibrary.wiley.com/doi/10.1002/jcb.27639

- https://www.mdpi.com/2073-4409/9/6/1375/htm

In your revision ensure you cite all your sources (including your own works), and quote or rephrase any duplicated text outside the methods section. Further consideration is dependent on these concerns being addressed.

Reviewers' comments:

Reviewer's Responses to Questions

**Comments to the Author**

1. Is the manuscript technically sound, and do the data support the conclusions?

Reviewer #1: Yes

Reviewer #2: No

2. Has the statistical analysis been performed appropriately and rigorously? 

Reviewer #1: No

Reviewer #2: No

3. Have the authors made all data underlying the findings in their manuscript fully available?

Reviewer #1: No

Reviewer #2: Yes

4. Is the manuscript presented in an intelligible fashion and written in standard English?

Reviewer #1: No

Reviewer #2: Yes

5. Review Comments to the Author

Reviewer #1: Luo, et. al. uses BPA in the drinking water of young male mice to investigate how this compound affects learning and memory impairment by exploring changes in microRNA expressed in hippocampus. Behavioral changes were observed using the Morris maze test. They report alterations in microRNAs that target genes related to modulation of synaptic plasticity, among other pathways. Some proteins related to synaptic plasticity, which were predicted targets of regulation of the altered microRNAs found, were downregulated in animals exposed to BPA. Although the subject is relevant, this work lacks some critical methodological and result information that needs to be addressed. Also, some more proteins could be analyzed, including glutamatergic receptors.

Major concerns:

• In lines 50 and 51 the phrase “In 2015, the global consumption of BPA reached 7.7 million metric tons” is also found in another paper (https://doi.org/10.1016/j.envint.2019.01.059 ). It is mandatory to rewrite and include this citation.

• In line 78 and 79 include the references of the studies.

• There is plenty of literature describing the harmful effects of BPA and information about the underlying mechanisms. Please add more details about the deleterious effects of BPA overexposure and the already known targets (i.e. estrogen receptors).

• Please check the writing and revise the grammar (i.e. change long term potentiation or depression for long-term potentiation and long-term depression).

• It is recommended to standardize the writing text adopting a space between number and the respective unit and correct possible mistakes (i.e. line 150, “3μg” change for “3 μg”; line 203 in “4C” include 4 °C).

• Information regarding the animal’s age during and after the treatment is not clear. Data regarding animal age during the BPA treatment (start - end); and when the animals were submitted to behavior test and hippocampal extraction is missing. Also, the data about the water consumption and the weight recorded every two days is not reported. This information helps to clarify the effects of the treatment on the animal’s water consumption.

• Please, include the full name of the animal strain and the number of animals used to compound the four described groups. There is no information about which time dark/light cycle initiate-end. Also, it is not specified if all animal were submitted to behavior tests and/or hippocampal extraction.

• In the section “2.3 Hippocampus tissue collections” of materials and methods, it is necessaire to describe whether the hippocampus was collected bilaterally and how the mentioned coordinates were based (i.e. Bregma). Please let it clear whether other brain regions could be collected together with the hippocampus by this method.

• In the section “2.2 Tests of animal learning and memory ability” it is not indicating each time (or period) of the day the experiments were conducted. Characteristics of the maze and the environment is not disclosed (i.e. size of the maze, size of the run, visual clues, etc). Also it is not described how the result was computed since each training day has 4 section of 60 s.

• The section “2.4 miRNA library construction and sequencing” is not informed the method of RNA extraction and the exact number of sequenced microRNA.

• In the section “2.5 Bioinformatic evaluation” It is necessary to include more details about the analyses. If it is possible, the author could indicate the result of each step from the evaluation procedure that could complement the figure 2.

• In the section “2.7 Western blot analyses” the catalog number of the antibodies used is not reported. It is described that primary antibodies were made in rabbit and secondary antibodies were goat anti-mouse IgG. Please check this information. Also, there is a lack of information regarding the method of analysis and the normalizing procedure. It is necessary to disclose the entire blot membranes, not just the area where the bands are.

• In the result section and the figure legends are not informing the values and the statistical analysis adopted for each analyses (i.e. mean value, standard error, etc).

• Many figures' information is hard to visualize due to the small font size adopted in some cases.

• In the section “3.5 BPA affected regulation of LTP and LTD regulatory miRNAs’ targets involved in neurotoxcity”, change “neurotoxciy” for neurotoxicity. In line 277 it says that 8 microRNAs are shown in the table 3 associated with LTP and LTD, but only seven microRNAs is reported for LPT. In the line 280-281 “In addition, miR-10b-5p and regulate LTP not LTD.” It seems that some information is missing. Also, it requires a full description of microRNA modulation on their targets in the legend or in this result section.

• In line 292 change “gorithm” for algorithm

• In line 314 correct the “neutoxicant”.

• In line 319, include citation about the impact of BPA on microRNA expression (i.e. https://doi.org/10.1002/jat.4025)

• In line 321, correct “major control center” for a more appropriate term such as brain region related to learning and memory.

• In line 329 change “neurobehaviors” for a more appropriate term such as cognitive process.

• In line 338-340 correct the sentence “Characteristic changes in synapses that occur LTP or LTD cannot be accounted for by global upregulation of translation.”

• The conclusion is too long and seems like a summary of the result section. It should bring the hypothesis and the main results that supported or not it.

• The conclusion is very long and looks like a summary of the results section. The conclusion must bring the hypothesis and the main results that support it or not. In addition, this session may contain possibilities raised from the main findings.

Reviewer #2: In this manuscript the authors assess whether delivery of BPA in drinking water can affect learning and memory (as assessed via the Morris Water Maze) and microRNA expression (as assessed by smallRNA-Seq and qRT-PCR) and microRNA target levels (via western blot). The authors show that BPA impairs learning at the highest tested dose and they find that there are associated changes in microRNA.

The work is not rigourously presented or interpreted. The bioinformatics are not presented in a systematic manner and are not convincing. Little effort is taken to draw in existing literature regarding alteration in microRNA following LTP-induction. Little consideration is taken as to the direction of change and how that might help explain the biology in focus here. The Discussion is weak and does not shed a great deal of new light on this field. The authors have used RNA-Seq of whole tissue. The field has really moved away from such analyses. The authors miss the opportunity to interpret their microRNA findings with mRNA data which could have been derived from these samples (or perhaps exist in the literature).

Abstract:

“…that regulate the expression and degradation of proteins.” Incorrect

“Seventeen miRNAs were significantly regulated by BPA.” By what test?

“Bioinformatic analysis of GOTERM_BP_FAT” What does this mean?

“Protein levels of CaMKII,…” Why?

“novel miRNA biomarkers” in what way?

There is no mention of the MWM data

Methods:

Include a justification of the doses of BPA used.

“Total RNA was isolated” how?

“purity was evaluated using the NanoPhotometer®…” What parameters were considered acceptable?

“(a p-value < 0.05 is recommended)” is this what was used?

2.5 Bioinformatic evaluation: this section is not written with clarity- rewrite

“and the final targets were integrated from at least two different programs” How were they chosen? What systematic approach was taken?

Provide the RT-PCR methods in standard format (not as a recipe)

Define: CaMKII, MEK1/2, IP3R and PLCβ4 and other abbreviations.

ß-actin is not a suitable control as actin dynamics have been shown numerous times to contribute to synaptic plasticity.

Why were nonreducing conditions used for the western blot analyses?

Provide the catalogue numbers of the Abcam, USA antibodies.

Statistical Section: How was multiple testing controlled for?

Explicitly state the ‘n’ used in each experiment and whether the RNA-Seq and qRT-PCR were carried out using the same tissue.

Results:

“the total number of crossings over the platform and time spent in the target quadrant in

5mg/kg BPA exposed group was significantly shorter compared with the control group (Fig.1C and D). Correct the syntax here.

Can the degree of impairment be correlated with the alteration in microRNA?

3.2: should explain how the data were generated. There is no indication of standard deviation/error

Fig.2 Legend should describe the statistical approach used.

3.4: “To gain a better understanding…We list the significantly changed GO terms (GOTERM BP FAT) related to learning or memory ” Were GO terms related to other pathways identified? This should be exploratory, but seems to be used to forefeel the authors hypothesis. If this is hypothesis driven research, then the authors should explore the existing literature of LTP and LTD-related microRNA and test whether these are regulated in their model.

As the same microRNA are grouped under the LTD and LTP banner, some consideration to the direction of change should be given.

Fig.3: “three independent experiments” Are these the same animals from the smallRNA-Seq work?

“The target genes involved in LTP and LTD were screened out in Fig. 4, which were predicted with at least 2 different algorithms.” This sentence is not clear.

Define ‘AMPARs’ etc

Fig.4: what is the source of this figure? How were miRs ‘screened’?

For example, AMPAR was regulated by miR-96-5p,

miR-182-5p and miR-125a-3p and miR-24-3p, CaMKIV was regulated

by miR-182-5p, miR-96-5p and miR-10b-3p, CaMKII was regulated by

miR-24-3p and miR-10b-5p, MEK1/2 was reglulated by miR-96-5p and

miR-182-5p, etc.

Fig.5: B’actin control is extremely overexposed. These type of data are unusable. This image must be replaced by one taken in the linear range of the film and the data reassessed.

Discussion: weak, not insightful

Consider how BPA might regulate microRNA expression.

6. PLOS authors have the option to publish the peer review history of their article (what does this mean?). If published, this will include your full peer review and any attached files.

Reviewer #1: **Yes: **Guilherme Shigueto Vilar Higa

Reviewer #2: **Yes: **Joanna Williams

---

## [Author Response · Author response to Decision Letter 0]

30 Jul 2022

The responses to the academic editor’ comments (Journal Requirements) are as following:

Journal Requirements:

Comment 1: Please ensure that your manuscript meets PLOS ONE's style requirements, including those for file naming. The PLOS ONE style templates can be found at

Response: Thank you for your helpful suggestion. We have modified our format according to PLOS ONE's style requirements.

Comment 2: Please include your tables as part of your main manuscript and remove the individual files. Please note that supplementary tables (should remain/ be uploaded) as separate "supporting information" files

Response: Thank you for your helpful suggestion. We have included our tables as part of our main manuscript and removed the individual files, and noted that supplementary tables should be uploaded as separate "supporting information" files.

Comment 3: Please include your full ethics statement in the ‘Methods’ section of your manuscript file. In your statement, please include the full name of the IRB or ethics committee who approved or waived your study, as well as whether or not you obtained informed written or verbal consent. If consent was waived for your study, please include this information in your statement as well.

Response: Thank you for your helpful suggestion. We have included our full ethics statement in the ‘Methods’ section of our manuscript.

Comment 4: PLOS requires an ORCID iD for the corresponding author in Editorial Manager on papers submitted after December 6th, 2016. Please ensure that you have an ORCID iD and that it is validated in Editorial Manager. To do this, go to ‘Update my Information’ (in the upper left-hand corner of the main menu), and click on the Fetch/Validate link next to the ORCID field. This will take you to the ORCID site and allow you to create a new iD or authenticate a pre-existing iD in Editorial Manager. Please see the following video for instructions on linking an ORCID iD to your Editorial Manager account: https://www.youtube.com/watch?v=_xcclfuvtxQ.

Response: Thank you for your helpful suggestion. We ensure that we have an ORCID iD and it is validated in Editorial Manager.

Comment 5: In your Data Availability statement, you have not specified where the minimal data set underlying the results described in your manuscript can be found. PLOS defines a study's minimal data set as the underlying data used to reach the conclusions drawn in the manuscript and any additional data required to replicate the reported study findings in their entirety. All PLOS journals require that the minimal data set be made fully available. For more information about our data policy, please see http://journals.plos.org/plosone/s/data-availability.

Response: Thank you for your helpful suggestion. All relevant data and the minimal data set are available in Supporting Information files, and our Data Availability statement has been provided in our cover letter.

Comment 6: We note that the grant information you provided in the ‘Funding Information’ and ‘Financial Disclosure’ sections do not match.

Response: Thank you for your helpful suggestion. We ensure that we provide the correct grant numbers.

Comment 7: Thank you for stating the following financial disclosure:

“NO”

Response: Thank you for your helpful suggestion. This work was supported by grants from the National Natural Science Foundation of China (No. 81273038). The funder is Fengyuan Piao. He took part in conceiving and designing this experiments. We have added this information in our cover letter.

Comment 8: We noticed you have some minor occurrence of overlapping text with the following previous publication(s), which needs to be addressed:

- https://onlinelibrary.wiley.com/doi/10.1002/jcb.27639

- https://www.mdpi.com/2073-4409/9/6/1375/htm

In your revision ensure you cite all your sources (including your own works), and quote or rephrase any duplicated text outside the methods section. Further consideration is dependent on these concerns being addressed.

Response: Thank you for your helpful suggestion. We have cited all our sources (including our own works), and quoted or rephrased any duplicated text outside the methods section.

Reviewers' comments:

Reviewer's Responses to Questions

Comments to the Author

1. Is the manuscript technically sound, and do the data support the conclusions?

Reviewer #1: Yes

Reviewer #2: No

2. Has the statistical analysis been performed appropriately and rigorously?

Reviewer #1: No

Reviewer #2: No

3. Have the authors made all data underlying the findings in their manuscript fully available?

Reviewer #1: No

Reviewer #2: Yes

4. Is the manuscript presented in an intelligible fashion and written in standard English?

Reviewer #1: No

Reviewer #2: Yes

5. Review Comments to the Author

Reviewer #1: Luo, et al., uses BPA in the drinking water of young male mice to investigate how this compound affects learning and memory impairment by exploring changes in microRNA expressed in hippocampus. Behavioral changes were observed using the Morris maze test. They report alterations in microRNAs that target genes related to modulation of synaptic plasticity, among other pathways. Some proteins related to synaptic plasticity, which were predicted targets of regulation of the altered microRNAs found, were downregulated in animals exposed to BPA. Although the subject is relevant, this work lacks some critical methodological and result information that needs to be addressed. Also, some more proteins could be analyzed, including glutamatergic receptors.

Response: Thank you very much for reviewing our manuscript and giving us so many useful suggestions. Your comments have helped us to improve our article significantly. We highlighted all changes in yellow in our revised manuscript. In addition, protein analysis of glutamatergic receptor was added in the part of western blot test.

Major concerns:

Comment 1: In lines 50 and 51 the phrase “In 2015, the global consumption of BPA reached 7.7 million metric tons” is also found in another paper (https://doi.org/10.1016/j.envint.2019.01.059 ). It is mandatory to rewrite and include this citation.

Response: Thank you for your helpful suggestion. We have rewrote this sentence and included the citation of Wang et al.

Comment 2: In line 78 and 79 include the references of the studies.

Response: Thank you for your helpful suggestion. We have added the references of the studies.

Comment 3: There is plenty of literature describing the harmful effects of BPA and information about the underlying mechanisms. Please add more details about the deleterious effects of BPA overexposure and the already known targets (i.e. estrogen receptors).

Response: Thank you for your helpful suggestion. We have added more details about the deleterious effects of BPA overexposure and the already known targets.

Comment 4: Please check the writing and revise the grammar (i.e. change long term potentiation or depression for long-term potentiation and long-term depression).

Response: Thank you for your helpful suggestion. We have checked the writing and revise the grammar.

Comment 5: It is recommended to standardize the writing text adopting a space between number and the respective unit and correct possible mistakes (i.e. line 150, “3μg” change for “3 μg”; line 203 in “4C” include 4 °C).

Response: Thank you for your helpful suggestion. We have standardized our writing text.

Comment 6: Information regarding the animal’s age during and after the treatment is not clear. Data regarding animal age during the BPA treatment (start - end); and when the animals were submitted to behavior test and hippocampal extraction is missing. Also, the data about the water consumption and the weight recorded every two days is not reported. This information helps to clarify the effects of the treatment on the animal’s water consumption.

Response: Thank you for your helpful suggestion. BPA treatment started at the 4 weeks of mice, and ended after 8 weeks administration. After 8 weeks of BPA treatment, the morris water maze (MWM) test was performed. After MWM test, mice were euthanized and hippocampus tissues were collected. 

We are sorry that we measure weight of mice every week, but we wrote it every two days. We have revised this information in the revised manuscript: The volumes of water consumption were measured every 2 days, the weights of the mice were measured every 1 week, and there were no significant differences in water consumption and the body weights of the mice between the BPA-exposed and control groups. In addition, the information of water consumption and weight was added in the revised manuscript (Fig 1).

Comment 7: Please, include the full name of the animal strain and the number of animals used to compound the four described groups. There is no information about which time dark/light cycle initiate-end. Also, it is not specified if all animal were submitted to behavior tests and/or hippocampal extraction.

Response: Thank you for your helpful suggestion. We have added the full name of the animal strain (Kunming mice) and the number of mice used in four groups (n = 20 in each group). The time dark/light cycle initiate at 7:00 a.m. and end at 7:00 p.m. After 8 weeks of BPA treatment, the morris water maze (MWM) test was performed in all mice from each group. After MWM test, mice were euthanized and hippocampus tissue were collected. We have added these information in our revised manuscript.

Comment 8: In the section “2.3 Hippocampus tissue collections” of materials and methods, it is necessaire to describe whether the hippocampus was collected bilaterally and how the mentioned coordinates were based (i.e. Bregma). Please let it clear whether other brain regions could be collected together with the hippocampus by this method.

Response: Thank you for your helpful suggestion. Hippocampal CA1 samples were collected from the both side according to the following coordinates relative to the bregma: −2 mm at the anterior/posterior axis, ±1.8 mm at the lateral/medial axis and −1.5 mm at the dorsal/ventral axis. We have added this description in our revised manuscript.

Comment 9: In the section “2.2 Tests of animal learning and memory ability” it is not indicating each time (or period) of the day the experiments were conducted. Characteristics of the maze and the environment is not disclosed (i.e. size of the maze, size of the run, visual clues, etc). Also it is not described how the result was computed since each training day has 4 section of 60 s.

Response: Thank you for your helpful suggestion. We have added the period of the day the MWM tests, the characteristics of the maze and the environment of the experiment in our revised manuscript. The mean value of the time that mice needed to find the platform (4 sections of starting at different quadrants of the pool) was defined as the escape latency, so the data presented is the daily average of four trials.

Comment 10: The section “2.4 miRNA library construction and sequencing” is not informed the method of RNA extraction and the exact number of sequenced microRNA.

Response: Thank you for your helpful suggestion. We have added method of RNA extraction in our method part and the exact number of sequenced microRNA in S1 Table.

Comment 11: In the section “2.5 Bioinformatic evaluation” It is necessary to include more details about the analyses. If it is possible, the author could indicate the result of each step from the evaluation procedure that could complement the figure 2.

Response: Thank you for your helpful suggestion. We have added more details about our analyses, including the steps of evaluation for complementing the figure 2.

Comment 12: In the section “2.7 Western blot analyses” the catalog number of the antibodies used is not reported. It is described that primary antibodies were made in rabbit and secondary antibodies were goat anti-mouse IgG. Please check this information. Also, there is a lack of information regarding the method of analysis and the normalizing procedure. It is necessary to disclose the entire blot membranes, not just the area where the bands are.

Response: Thank you for your helpful suggestion. We are sorry for our mistakes. We have corrected the information of primary and secondary antibodies. Information regarding the method of analysis and the normalizing procedure have been added. The entire blot membranes of all proteins were provided in supplementary data. 

Comment 13: In the result section and the figure legends are not informing the values and the statistical analysis adopted for each analyses (i.e. mean value, standard error, etc).

Response: Thank you for your helpful suggestion. We have added the values and the statistical analysis in the result section and figure legends.

Comment 14: Many figures' information is hard to visualize due to the small font size adopted in some cases.

Response: Thank you for your helpful suggestion. We have checked all of our figures and corrected them more clearly. 

Comment 15: In the section “3.5 BPA affected regulation of LTP and LTD regulatory miRNAs’ targets involved in neurotoxcity”, change “neurotoxciy” for neurotoxicity. In line 277 it says that 8 microRNAs are shown in the table 3 associated with LTP and LTD, but only seven microRNAs is reported for LPT. In the line 280-281 “In addition, miR-10b-5p and regulate LTP not LTD.” It seems that some information is missing. Also, it requires a full description of microRNA modulation on their targets in the legend or in this result section.

Response: Thank you for your helpful suggestion. We are sorry for us mistake of the miRNA’s amount and us inaccurate use of words. We have already corrected them in our revised manuscript. The full description of microRNA modulation on their targets have been added in Table 4.

Comment 16: In line 292 change “gorithm” for algorithm

Response: Thank you for your helpful suggestion. We have corrected the word.

Comment 17: In line 314 correct the “neutoxicant”.

Response: Thank you for your helpful suggestion. We have corrected the word.

Comment 18: In line 319, include citation about the impact of BPA on microRNA expression (i.e. https://doi.org/10.1002/jat.4025)

Response: Thank you for your helpful suggestion. We have cited this reference.

Comment 19: In line 321, correct “major control center” for a more appropriate term such as brain region related to learning and memory.

Response: Thank you for your helpful suggestion. We have corrected the word.

Comment 20: In line 329 change “neurobehaviors” for a more appropriate term such as cognitive process.

Response: Thank you for your helpful suggestion. We have changed the word.

Comment 21: In line 338-340 correct the sentence “Characteristic changes in synapses that occur LTP or LTD cannot be accounted for by global upregulation of translation.”

Response: Thank you for your helpful suggestion. We have corrected the sentence.

Comment 22: The conclusion is too long and seems like a summary of the result section. It should bring the hypothesis and the main results that supported or not it.

Response: Thank you for your helpful suggestion. We have rewrote our conclusion in the revised manuscript, and we hope it gets better.

Comment 23: The conclusion is very long and looks like a summary of the results section. The conclusion must bring the hypothesis and the main results that support it or not. In addition, this session may contain possibilities raised from the main findings.

Response: Thank you for your helpful suggestion. That’s so kind of you for helping us to write the conclusion. We have tried to write it again, and we hope it gets better.

Reviewer #2: In this manuscript the authors assess whether delivery of BPA in drinking water can affect learning and memory (as assessed via the Morris Water Maze) and microRNA expression (as assessed by smallRNA-Seq and qRT-PCR) and microRNA target levels (via western blot). The authors show that BPA impairs learning at the highest tested dose and they find that there are associated changes in microRNA.

The work is not rigourously presented or interpreted. The bioinformatics are not presented in a systematic manner and are not convincing. Little effort is taken to draw in existing literature regarding alteration in microRNA following LTP-induction. Little consideration is taken as to the direction of change and how that might help explain the biology in focus here. The Discussion is weak and does not shed a great deal of new light on this field. The authors have used RNA-Seq of whole tissue. The field has really moved away from such analyses. The authors miss the opportunity to interpret their microRNA findings with mRNA data which could have been derived from these samples (or perhaps exist in the literature).

Response: Thank you very much for reviewing our manuscript and giving us so many useful suggestions. In our new manuscript, we have corrected all the issues according to your suggestions. We are sorry for us inaccurate description, the tissue we used for RNA-Seq is hippocampus CA1 because we take the material according to its anatomical position (Please see “Hippocampus tissue collection” in materials and methods). More details on bioinformatics information were presented, such as the change direction of the differentially expressed miRNAs, significantly changed GO terms and miRNAs’ target genes linked to LTP and LTD. And we have tried to discuss more and deeper about the miRNA and synaptic plasticity. We hope the revised manuscript would be better. 

Abstract

Comment 1::“…that regulate the expression and degradation of proteins.” Incorrect

Response: Thank you for your helpful suggestion. We corrected to “miRNAs control physiological and pathological processes by inhibiting translation, and promoting mRNA degradation”.

Comment 2: “Seventeen miRNAs were significantly regulated by BPA.” By what test?

Response: Thank you for your question. We analyzed the impacts of BPA on miRNA expression profile by high-throughput sequencing in mice hippocampus, seventeen miRNAs were significantly differentially expressed after BPA exposure, of these, 13 and 4 miRNAs were up- and downregulated, respectively.

Comment 3: “Bioinformatic analysis of GOTERM_BP_FAT” What does this mean?

Response: Thank you for your question. We are sorry for us inaccurate use of terms, we have corrected “GOTERM_BP_FAT” to “Gene Ontology (GO)”. 

Comment 4: “Protein levels of CaMKII,…” Why?

Response: Thank you for your question. The expression levels of proteins of five target genes (CaMKII, MEK1/2, IP3R, AMPAR1 and PLCβ4) were investigated via western blot, for further verifying the results of gene target analysis.

Comment 5: “novel miRNA biomarkers” in what way?

Response: Thank you for your question. This study provides valuable information for novel miRNA biomarkers to detect changes in impairment of learning and memory induced by BPA exposure.

Comment 5: There is no mention of the MWM data.

Response: Thank you for your helpful suggestion. We have added the MWM data in our revised abstract.

Methods:

Comment 6: Include a justification of the doses of BPA used.

Response: Thank you for your helpful suggestion. The justification of the doses of BPA used were included in our revised abstract.

Comment 7: “Total RNA was isolated” how?

Response: Thank you for your helpful suggestion. Total RNA was extracted from hippocampal CA1 tissues by using RNAiso Plus according to the manufacturer’s instruction (Takara, Japan). We have corrected in our revised manuscript.

Comment 8: “purity was evaluated using the NanoPhotometer®…” What parameters were considered acceptable?

Response: Thank you for your helpful suggestion. We are sorry for us inaccurate use of words. Total RNA was quantified with the NanoPhotometer® spectrophotometer (IMPLEN, CA, USA). Only RNA samples with an A260/A280 of 1.8–2.2 were employed for reverse transcription. We have corrected in our revised manuscript.

Comment 9: “(a p-value < 0.05 is recommended)” is this what was used?

Response: Thank you for your question. We are sorry for us unclear descriptions. We have rewrote this part in our revised bioinformatic evaluation paragraph.

We selected the differentially expressed miRNAs according to the log2 (FC) and p value threshold. |log2 (FC) |≥ 1 and p value < 0.05 was considered as significant difference. 

Comment 10: Bioinformatic evaluation: this section is not written with clarity- rewrite

“and the final targets were integrated from at least two different programs” How were they chosen? What systematic approach was taken?

Response: Thank you for your helpful suggestion. We are sorry for us unclear decrciption. We have rewrote the section of bioinformatic evalution. The targets of miRNAs were selected by the following two steps: 1. Potential target genes of all the differentially expressed miRNAs were predicted from data in the databases: miRDB, miRanda, miRWalk, TargetScan, DIANA-mirPath, and miRNA.org. 2. The results intersected from at least two different programs were retained as the final set of target genes. 

Comment 11: Provide the RT-PCR methods in standard format (not as a recipe)

Response: Thank you for your helpful suggestion. We have tried to provide the Real Time-PCR methods in standard format, and we hope it gets better.

Comment 12: Define: CaMKII, MEK1/2, IP3R and PLCβ4 and other abbreviations.

Response: Thank you for your helpful suggestion. We have defined all the abbreviations in S2 Table and Fig 5 legend.

Comment 13: ß-actin is not a suitable control as actin dynamics have been shown numerous times to contribute to synaptic plasticity.

Response: Thank you for your helpful suggestion. The GADPH has been used instead of ß-actin in our revised manuscript.

Comment 14: Why were nonreducing conditions used for the western blot analyses? Provide the catalogue numbers of the Abcam, USA antibodies.

Response: Thank you for your question. If some antibody target sites contain disulfide bond, the reductant condition will open the disulfide bond and become sulfhydryl and such antibodies will not be recognized. Therefore, nonreducing conditions were used for the western blot analyses. The catalogue numbers have been provided in our revised manuscript.

Comment 15: Statistical Section: How was multiple testing controlled for?

Explicitly state the ‘n’ used in each experiment and whether the RNA-Seq and qRT-PCR were carried out using the same tissue.

Response: Thank you for your helpful suggestion. Prior to analyses, we confirmed data normality and homogeneity of variances by D’Agostino–Pearson and Barlett’s tests, respectively. Difference among each group was analyzed by one-way analysis of variance (ANOVA), followed by the Tukey’s multiple-comparison tests. 

We have stated the ‘n’ used in each experiment. The RNA from the hippocampus of mice (three from control group and three from 5mg/mg BPA group) were used to construct miRNA expression profile by RNA-Seq (n=3). The RNA from the hippocampus of the other 6 mice in each group were used to verify the miRNA expression by Real Time-PCR (n=6). Therefore, we have added this information in Materials part and figure legend.

Results:

Comment 16: “the total number of crossings over the platform and time spent in the target quadrant in 5mg/kg BPA exposed group was significantly shorter compared with the control group (Fig.1C and D). Correct the syntax here.

Response: Thank you for your helpful suggestion. We have corrected the syntax in the revised manuscripts.

Comment 17: Can the degree of impairment be correlated with the alteration in microRNA?

Response: Thank you for your question. In our study, the impairment of learning and memory was shown in mice exposed at 5mg/kg BPA, and no significant change in mice exposed at 0.05 and 0.5 mg/kg BPA. However, the alteration of some miRNAs was not only found in mice hippocampus which exposed at 5mg/kg BPA, but also found in 0.5 mg/kg BPA exposure group. It may be that BPA exposure caused genetic changes first, and then behavioral changes, which we need to explore in the future.

Comment 18: should explain how the data were generated. There is no indication of standard deviation/error

Response: Thank you for your helpful suggestion. We have explained how the data were generated and added the indication of standard error.

Comment 19: Fig.2 Legend should describe the statistical approach used.

Response: Thank you for your helpful suggestion. We selected the differentially expressed miRNAs according to the log2 (FC) and p value threshold. |log2 (FC) |≥ 1 and p value < 0.05 was considered as significant difference. We have described the statistical approach used in figure legend. 

Comment 20: “To gain a better understanding…We list the significantly changed GO terms (GOTERM BP FAT) related to learning or memory” Were GO terms related to other pathways identified? This should be exploratory, but seems to be used to forefeel the authors hypothesis. If this is hypothesis driven research, then the authors should explore the existing literature of LTP and LTD-related microRNA and test whether these are regulated in their model.

Response: Thank you for your helpful suggestion. We are sorry for using our hypothesis here (because there are more than 20 GO terms significantly changed, we selected the terms). We listed out the top 10 GO terms in our revised manuscript as following table:

Term p-Value Ontology

Protein phosphorylation

8.7E-5 Biological process

Signal transduction

9.8E-5 Biological process

Learning or memory

1.1E-4 Biological process

Learning

1.3E-4 Biological process

Chemical synaptic transmission

1.4E-4 Biological process

Regulation of synaptic plasticity 

1.7E-4 Biological process

MAPK cascade

2.9E-4 Biological process

Neurological system process

3.7E-4 Biological process

Negative regulation of neuron apoptotic process

6.6E-4 Biological process

Locomotory behavior

8.6E-4 Biological process

Comment 21: As the same microRNA are grouped under the LTD and LTP banner, some consideration to the direction of change should be given.

Response: Thank you for your helpful suggestion. Among the miRNAs are under the LTP and LTD, two of them are downregulated, six of them are upregulated by BPA treatment. We have added the change direction of them in Table 3 in our revised manuscript. 

Comment 22: Fig.3: “three independent experiments” Are these the same animals from the smallRNA-Seq work?

Response: Thank you for your helpful suggestion. We are sorry for us unclear statement. We meant to say “n= 6 from three independent experiments and performed in triplicates”. These are not the same animals from the smallRNA-Seq work. We have corrected and clarified it in our revised manuscript.

Comment 23: “The target genes involved in LTP and LTD were screened out in Fig. 4, which were predicted with at least 2 different algorithms.” This sentence is not clear.

Response: Thank you for your helpful suggestion. We are sorry for us unclear statement. For minimizing the number of putative and maybe false positive targets, we select intersections from at least two different databases as miRNA’s targets. 

Comment 24: Define ‘AMPARs’ etc

Response: Thank you for your helpful suggestion. We have defined ‘AMPARs’ and other abbreviations in the legend of Fig.4.

Comment 25:Fig.4: what is the source of this figure? How were miRs ‘screened’? For example, AMPAR was regulated by miR-96-5p, miR-182-5p and miR-125a-3p and miR-24-3p, CaMKIV was regulated by miR-182-5p, miR-96-5p and miR-10b-3p, CaMKII was regulated by miR-24-3p and miR-10b-5p, MEK1/2 was reglulated by miR-96-5p and miR-182-5p, etc.

Response: Thank you for your helpful suggestion. The pathway is obtained from KEGG pathway and confirmed through an NCBI PubMed. The miRNAs was screened by miRNAs-target predictional algorithms, including miRanda, miRWalk, miRDB, TargetScan, miRNA.org and DIANA-mirPath. For minimizing the number of putative and maybe false positive targets, we select intersections from at least two different databases as miRNA’s targets for minimizing the number of putative and maybe false positive targets. In Table 4, it was shown the detailed information of target genes of miRNA.

The functional regulation pathway from BPA exposure to impairment of learning and memory was summarized using LTP and LTD-related miRNAs and their predicted target genes in Fig.5.

Comment 26: Fig.5: ß-actin control is extremely overexposed. These type of data are unusable. This image must be replaced by one taken in the linear range of the film and the data reassessed.

Response: Thank you for your helpful suggestion. We have used GAPDH as control instead of ß-actin, therefore the data reassessed in our revised version.

Comment 27: Discussion: weak, not insightful

Consider how BPA might regulate microRNA expression.

Response: Thank you for your helpful suggestion. We have rewrote the part of discussion about how BPA might regulate microRNA expression in our revised manuscript and marked it yellow. We hope it gets better.

6. PLOS authors have the option to publish the peer review history of their article (what does this mean?). If published, this will include your full peer review and any attached files.

Do you want your identity to be public for this peer review? For information about this choice, including consent withdrawal, please see our Privacy Policy.

Reviewer #1: Yes: Guilherme Shigueto Vilar Higa

Reviewer #2: Yes: Joanna Williams

---

## [Decision Letter · Decision Letter 1]

20 Sep 2022

PONE-D-22-07675R1Long term potentiation and depression regulatory microRNAs were highlighted in Bisphenol A induced learning and memory impairment by microRNA sequencing and bioinformatics analysisPLOS ONE

Dear Dr. Zhang,

Thank you for submitting your manuscript to PLOS ONE. After careful consideration, we feel that it has merit but does not fully meet PLOS ONE’s publication criteria as it currently stands. Therefore, we invite you to submit a revised version of the manuscript that addresses the points raised during the review process.

Please,  carefully consider all comments from reviewer. 

We look forward to receiving your revised manuscript.

Kind regards,

Alexandre Hiroaki Kihara, Ph.D.

Academic Editor

PLOS ONE

Reviewers' comments:

Reviewer's Responses to Questions

**Comments to the Author**

1. If the authors have adequately addressed your comments raised in a previous round of review and you feel that this manuscript is now acceptable for publication, you may indicate that here to bypass the “Comments to the Author” section, enter your conflict of interest statement in the “Confidential to Editor” section, and submit your "Accept" recommendation.

Reviewer #1: (No Response)

2. Is the manuscript technically sound, and do the data support the conclusions?

Reviewer #1: Yes

3. Has the statistical analysis been performed appropriately and rigorously? 

Reviewer #1: No

4. Have the authors made all data underlying the findings in their manuscript fully available?

Reviewer #1: No

5. Is the manuscript presented in an intelligible fashion and written in standard English?

Reviewer #1: (No Response)

6. Review Comments to the Author

Reviewer #1: The authors have addressed most of the previous comments. Unfortunately, some issues remain to be clarified.

(1) It is described in the section “Hippocampus tissue collection” of Material and Methods that tissue from coordinate -2 mm AP, ±1.8 mm M/L, and 1.5 DV was collected, which corresponds to a small portion of CA1 (sub pyramidal part of CA1 stratum radiatum). Please describe whether only this portion of CA1 was collected or other regions were also included. If the entire hippocampus were collected, it is not necessary to cite the coordinates. Please, make it clear.

(2) In the “Western Blot” section, it is not clear how the normalization by GAPDH was performed. Please describe if the GAPDH staining was performed in stripped membranes used to identify the target protein’s antibodies. Also, inform whether the normalization was performed by the ratio from band intensity of target proteins against GAPDH. The blotting membrane provided in the supplementary section seems to be cut and does not contain the entire sample used for quantification (only four bands for each target protein). Besides that, it does not present the standard protein ladder, which aids the identification of the right band to be quantified. Please, it is necessary to disclose the entire membrane with all samples used in the present work.

(3) In the “Statistical analysis”, it is described that data was submitted to homogeneity and normality test. If all data set follows a normal distribution, please indicate it in the text. If it is not the case, please report the statistical test employed.

(4) In figure 2A, please indicate which trial (i.e., trial 1-4) the representative swimming route belongs to and if it was obtained from the same animal.

(5) The miRNA analysis may have been impacted by the low number of samples employed in the analyzed groups, now indicated by the authors. It is possible that the change in miRNA expression after treatment with BPA was affected by the small size sample since some displaced miRNA indicated in the scatter plot were not significant in the statistical test. Please justify the small sample size for the groups.

(6) In the result section, please indicate in the text values of the mean and SEM represented in the graphs.

Minor issues

(1) Line 26-28 Change “Results showed that mice treated with BPA displayed impairments of spatial learning and memory, and the expression of miRNAs in hippocampus changed.” For “Results showed that mice treated with BPA displayed impairments of spatial learning and memory and changes in the expression of miRNAs in the hippocampus.”

(2) Line 99- change “BPA induced learning and memory..” for BPA-induced

(3) Line 101- idem

(4) Line 556-557- idem

(5) Line 211 – correct the word “normalrized”

(6) Line 213 – Correct the verb tense

(7) Line 253- Change “4 C for 1h” to “4 °C for 1 h”

(8) Line 271- Please include the antibody’s company full name and the catalog number of the product.

(9) In figure 3A, please insert the information in the legend about the blue dots.

(10) Line 427- Change “all of them protein levels..” for “all analyzed protein levels…”

(11) Line 447 changes “manipulation” to “involved in”

7. PLOS authors have the option to publish the peer review history of their article (what does this mean?). If published, this will include your full peer review and any attached files.

Reviewer #1: **Yes: **Guilherme Shigueto Vilar Higa

---

## [Author Response · Author response to Decision Letter 1]

14 Oct 2022

The responses to the reviewers’ comments are as following:

Review Comments to the Author

Reviewer #1: The authors have addressed most of the previous comments. Unfortunately, some issues remain to be clarified.

Response: Thank you very much for reviewing our manuscript and giving us so many useful suggestions. Your comments have helped us to improve our article significantly. We highlighted all changes in yellow in our revised manuscript.

Comment 1: It is described in the section “Hippocampus tissue collection” of Material and Methods that tissue from coordinate -2 mm AP, ±1.8 mm M/L, and 1.5 DV was collected, which corresponds to a small portion of CA1 (sub pyramidal part of CA1 stratum radiatum). Please describe whether only this portion of CA1 was collected or other regions were also included. If the entire hippocampus were collected, it is not necessary to cite the coordinates. Please, make it clear.

Response: Thank you for your helpful suggestion. We are sorry that we did not understand reviewer’s opinion well and wrote the location of hippocampus last time. In our study, the entire hippocampus was collected, because the hippocampus CA1 of mice was too small to collect. We removed the coordinates in this revised manuscript.

Comment 2：In the “Western Blot” section, it is not clear how the normalization by GAPDH was performed. Please describe if the GAPDH staining was performed in stripped membranes used to identify the target protein’s antibodies. Also, inform whether the normalization was performed by the ratio from band intensity of target proteins against GAPDH. The blotting membrane provided in the supplementary section seems to be cut and does not contain the entire sample used for quantification (only four bands for each target protein). Besides that, it does not present the standard protein ladder, which aids the identification of the right band to be quantified. Please, it is necessary to disclose the entire membrane with all samples used in the present work.

Response: Thank you for your helpful suggestion. In western blot, GAPDH staining was performed in stripped membranes used to identify the target protein’s antibodies, and normalization was performed by the ratio from band intensity of target proteins against GAPDH. We added this information in our revised manuscript.

We are sorry for that we didn’t present the standard protein ladder and disclose the entire membrane before (Our western blot were conducted with old developing equipment which could not merge the protein ladder and gray target bands, therefore we only took the photo of gray target bands, and cut the blotting membrane for saving the antibodies). We really value your suggestion, so we did the western blot again and used new developing equipment from other laboratories this time, and the data were afford in our revised supplement data.

Comment 3：In the “Statistical analysis”, it is described that data was submitted to homogeneity and normality test. If all data set follows a normal distribution, please indicate it in the text. If it is not the case, please report the statistical test employed.

Response: Thank you for your helpful suggestion. In our study, all data set fellows a normal distribution. We have added this information in our revised manuscript.

Comment 4：In figure 2A, please indicate which trial (i.e., trial 1-4) the representative swimming route belongs to and if it was obtained from the same animal.

Response: Thank you for your helpful suggestion. The representative swimming routes in figure 2A, we have indicated the representative swimming route of trial 1-4. The representative swimming routes belong to the same animal in each group. We have added this information in our revised manuscript.

Comment 5： The miRNA analysis may have been impacted by the low number of samples employed in the analyzed groups, now indicated by the authors. It is possible that the change in miRNA expression after treatment with BPA was affected by the small size sample since some displaced miRNA indicated in the scatter plot were not significant in the statistical test. Please justify the small sample size for the groups.

Response: Thank you for your helpful suggestion. As you suggestion, we performed the statistical analysis for the displaced miRNAs indicated in the scatter plot. All the displaced miRNAs were significantly changed, however, the P value varies from 0.042 to <0.001. Therefore, chip or sequencing is the first step to screen differentially expressed miRNAs, Real Time-PCR is used to verify the miRNA expression which screened out. In Table 1 of our revised manuscript, we added the p-value of miRNAs with statistical differences in expression between the BPA treatment and control groups.

Comment 6：In the result section, please indicate in the text values of the mean and SEM represented in the graphs.

Response: Thank you for your helpful suggestion. We have indicated the mean and SEM represented in miRNA analysis in Table 1 of our revised manuscript.

Minor issues

Comment 1: Line 26-28 Change “Results showed that mice treated with BPA displayed impairments of spatial learning and memory, and the expression of miRNAs in hippocampus changed.” For “Results showed that mice treated with BPA displayed impairments of spatial learning and memory and changes in the expression of miRNAs in the hippocampus.”

Response: Thank you for your helpful suggestion. We have corrected this sentence.

Comment 2: Line 99- change “BPA induced learning and memory..” for BPA-induced

Response: Thank you for your helpful suggestion. We have changed “BPA induced learning and memory..” for BPA-induced.

Comment 3: Line 101- idem

Response: Thank you for your helpful suggestion. We have added “-”.

Comment 4: Line 556-557- idem

Response: Thank you for your helpful suggestion. We have added “-”.

Comment 5: Line 211 – correct the word “normalrized”

Response: Thank you for your helpful suggestion. We have correct the word to “normalized”.

Comment 6: Line 213 – Correct the verb tense

Response: Thank you for your helpful suggestion. We have correct the verb tense. Comment 7: Line 253- Change “4 C for 1h” to “4 °C for 1 h”

Response: Thank you for your helpful suggestion. We have correct “4 C for 1h” to “4 °C for 1 h”.

Comment 8: Line 271- Please include the antibody’s company full name and the catalog number of the product.

Response: Thank you for your helpful suggestion. We have added the antibody’s company full name and the catalog number.

Comment 9: In figure 3A, please insert the information in the legend about the blue dots.

Response: Thank you for your helpful suggestion. We have insert the information in the legend about the blue dots.

Comment 10: Line 427- Change “all of them protein levels..” for “all analyzed protein levels…”

Response: Thank you for your helpful suggestion. We have Changed “all of them protein levels..” for “all analyzed protein levels…”

Comment 11: Line 447 changes “manipulation” to “involved in”

Response: Thank you for your helpful suggestion. We have changed “manipulation” to “involved in”.

---

## [Decision Letter · Decision Letter 2]

29 Nov 2022

Long term potentiation and depression regulatory microRNAs were highlighted in Bisphenol A induced learning and memory impairment by microRNA sequencing and bioinformatics analysis

PONE-D-22-07675R2

Dear Dr. Zhang,

We’re pleased to inform you that your manuscript has been judged scientifically suitable for publication and will be formally accepted for publication once it meets all outstanding technical requirements.

Kind regards,

Alexandre Hiroaki Kihara, Ph.D.

Academic Editor

PLOS ONE

Additional Editor Comments (optional):

Reviewers' comments:

Reviewer's Responses to Questions

**Comments to the Author**

1. If the authors have adequately addressed your comments raised in a previous round of review and you feel that this manuscript is now acceptable for publication, you may indicate that here to bypass the “Comments to the Author” section, enter your conflict of interest statement in the “Confidential to Editor” section, and submit your "Accept" recommendation.

Reviewer #1: All comments have been addressed

2. Is the manuscript technically sound, and do the data support the conclusions?

Reviewer #1: Yes

3. Has the statistical analysis been performed appropriately and rigorously? 

Reviewer #1: Yes

4. Have the authors made all data underlying the findings in their manuscript fully available?

Reviewer #1: Yes

5. Is the manuscript presented in an intelligible fashion and written in standard English?

Reviewer #1: Yes

6. Review Comments to the Author

Reviewer #1: The authors have addressed most of the previous comments. Procedures are clearly described in this version of the manuscript, making it adequate for publication.

However, some comments were not fully implemented in the new version of the manuscript. It is recommended to consider the suggestions to improve the presentation of the paper.

Specifics issues

1)Regarding comment 6, the authors have included the mean and SEM of miRNA analysis in table 1. I suggest including these values for all presented results, as recommended in the last revision.

2)The authors have provide the full manbrane pictures from representative western blot results whith the standard protein ladder. I suggest to include the molecular weight of each protein ladder band in the suplementar figure.

7. PLOS authors have the option to publish the peer review history of their article (what does this mean?). If published, this will include your full peer review and any attached files.

Reviewer #1: No

---

## [Editor Report · Acceptance letter]

23 Dec 2022

PONE-D-22-07675R2 

Long-term potentiation and depression regulatory microRNAs were highlighted in Bisphenol A induced learning and memory impairment by microRNA sequencing and bioinformatics analysis 

Dear Dr. Zhang:

I'm pleased to inform you that your manuscript has been deemed suitable for publication in PLOS ONE. Congratulations! Your manuscript is now with our production department. 

Kind regards, 

on behalf of

Dr. Alexandre Hiroaki Kihara 

Academic Editor

PLOS ONE